# Fully Immersive Virtual Reality-Based Cognitive Remediation for Adults with Psychosocial Disabilities: A Systematic Scoping Review of Methods Intervention Gaps and Meta-Analysis of Published Effectiveness Studies

**DOI:** 10.3390/ijerph20021527

**Published:** 2023-01-14

**Authors:** Alessandra Perra, Chiara Laura Riccardo, Valerio De Lorenzo, Erika De Marco, Lorenzo Di Natale, Peter Konstantin Kurotschka, Antonio Preti, Mauro Giovanni Carta

**Affiliations:** 1International PhD in Innovation Sciences and Technologies, Department of Mechanical Chemistry and Materials Engineering, University of Cagliari, 09042 Cagliari, Italy; 2Department of Medical Sciences and Public Health, University of Cagliari, 09042 Cagliari, Italy; 3Department of Clinical and Biological Sciences, University of Turin, 10126 Turin, Italy; 4PRoMIND Services for Mental Health, 00133 Rome, Italy; 5Azienda Sociosanitaria Ligure 2, Dipartimento di Salute Mentale e delle Dipendenze, 17100 Savona, Italy; 6IDEGO Digital Psychology Society, 00133 Rome, Italy; 7Department of General Practice, University Hospital Wuerzburg, D-97080 Wuerzburg, Germany; 8Department of Neuroscience, University of Turin, 10126 Turin, Italy

**Keywords:** virtual reality, cognitive remediation, mental health, recovery, psychiatric rehabilitation

## Abstract

Background: Cognitive Remediation (CR) programs are effective for the treatment of mental diseases; in recent years, Virtual Reality (VR) rehabilitation tools are increasingly used. This study aimed to systematically review and meta-analyze the published randomized controlled trials that used fully immersive VR tools for CR programs in psychiatric rehabilitation. We also wanted to map currently published CR/VR interventions, their methods components, and their evidence base, including the framework of the development intervention of CR in fully immersive VR. Methods: Level 1 of evidence. This study followed the PRISMA extension for Scoping Reviews and Systematic Review. Three electronic databases (Pubmed, Cochrane Library, Embase) were systematically searched, and studies were included if they met the eligibility criteria: only randomized clinical trials, only studies with fully immersive VR, and only CR for the adult population with mental disorders. Results: We found 4905 (database) plus 7 (manual/citation searching articles) eligible studies. According to inclusion criteria, 11 studies were finally reviewed. Of these, nine included patients with mild cognitive impairment, one with schizophrenia, and one with mild dementia. Most studies used an ecological scenario, with improvement across all cognitive domains. Although eight studies showed significant efficacy of CR/VR, the interventions’ development was poorly described, and few details were given on the interventions’ components. Conclusions: Although CR/VR seems to be effective in clinical and feasibility outcomes, the interventions and their components are not clearly described. This limits the understanding of the effectiveness and undermines their real-world implementation and the establishment of a gold standard for fully immersive VR/CR.

## 1. Introduction

The economic and health impact of mental disorders represents a worldwide problem of public health [1,2]. According to the Global Burden of Mental Disease, it is estimated that 13% of the global population has experienced an episode of a mental disorder [3,4]. Cognitive impairment is a fundamental component of mental disorders that contribute to psychosocial disabilities and limits the recovery process in psychiatric rehabilitation [5,6]. Cognitive deficits are present in most disorders such as psychosis, mood, anxiety, personality, and eating disorders [7,8,9,10,11], but in schizophrenia is a core symptom [12] and also approximately 60% of people with bipolar disorder have neurocognitive impairment [13,14]. In general, cognitive deficits negatively impact personal, social, and work-occupational functioning [10,15,16,17] and they are a barrier to achieving a good quality of life and independent living [18,19,20].

Among others, cognitive remediation programs (CR) and physical activity are psychiatric and neurological rehabilitation interventions that showed to be effective in improving cognition, as the main outcome [6,21,22,23,24], as well as improving social and personal functioning outcome [25,26,27]. At the same time, physical activity and CR were proven to be effective to prevent cognitive decline in healthy populations [28,29,30]. Cognitive, social, and personal functions are important mental health outcomes to achieve a real impact in daily life. Indeed, CR programs are interventions based on behavioral training that aims to improve cognitive domains (memory, attention, executive functions, social cognition, and metacognition) to obtaining the persistence of the cognitive strategies and their generalization in daily life [31]. CR was found to be effective in the treatment of cognitive deficits in people with psycho-social disorders such as psychosis, mood disorders, anxiety disorders, personality, and eating disorders [8,9,11,32,33,34,35,36]. There is also evidence of its effectiveness in neuropsychological disorders such as dementia, Mild Cognitive Impairment (MCI), and behavioral disorders [37,38]. Currently, the application methods of CR include a set of heterogeneous interventions: computerized, paper, in individual or group sets [39].

In the Digital Era, the mental health care system is exposed to a technological revolution [40]. The decreasing costs and increasing convenience and power of digital media and new technologies affect how we provide and access care [41]. One of the increasing technological instruments is an extension of games such as Virtual Reality (VR). VR includes some interactive video gaming, virtual environments, and, commonly, a multisensory experience to merge real and virtual worlds [42]. VR environments elicit a sense of presence and stimulate real-time cognitive, emotional, behavioral, and physiological responses to real-time situations [43]. This sense of being there like in a real-life situation has led the researcher to describe VR environments as ecologically valid [44]. For this reason, researchers and clinicians try to design realistic scenarios that can be used to assess and improve the individual daily skills to respond to the health needs of people who have experienced psychosocial disabilities [45,46]. Current research on the clinical use of VR software has led to positive results in cardiovascular, neurological, and musculoskeletal rehabilitation [47,48]. VR has garnered significant attention as a cost-effective tool for the treatment of mental health problems [40,49]. It is currently used for the assessment of cognitive and motor deficits, for the psychotherapy of anxiety and phobia disorders, and social skill improvement in psychosis disorders [50,51,52,53]. The CR interventions in fully immersive VR program are increasingly used [54] and to date, in the scientific literature, there is preliminary evidence for CR program with fully immersive VR used in the treatment of mental disorders [55,56,57]. Although there is some preliminary evidence, the quality of the studies is still low and there are also many points of uncertainty. The sources of concern are the variability of the methods (duration and frequency) [55], the variability in the level of virtual immersion [58], and the reference framework for the development of complex intervention [46]. The last one is a very important aspect in the development of rehabilitative interventions to achieve a real impact in daily life [46,55].

The first aim of this study is to investigate the published literature on the fully immersive VR used for the CR program in psychiatric rehabilitation treatments in adults to describe the different methods of intervention components and, furthermore, to highlight gaps in the framework of the development intervention of CR in fully immersive VR, to understand if there is an explanation of the hypothesis/outcomes/methods and coherence with the games used and cognitive domains. In general, our goal is to map the current state of knowledge and to identify gaps in the existing literature that merit further research [59]. To our knowledge, in the literature, there are three systematic reviews [55,56,57] that assessed the effectiveness of randomized clinical trials, and no meta-analysis has been carried out. For this reason, the second aim is to evaluate the effectiveness of the current evidence on the effect of clinical outcomes of CR programs with fully immersive VR in people with psycho-social disabilities with a meta-analysis.

## 2. Methods

### 2.1. Search Strategy

This study used an established scoping review methodology [60] and followed the reporting guidelines outlined in the PRISMA [61] and its extension for Scoping Reviews [62]. The articles to be included were identified through the keys words “Virtual Reality” [All Fields] AND “Mental Health” [All Fields]; “Virtual Reality” [All Fields] AND “Cognitive Remediation” [All Fields], “Virtual Reality” [All Fields] AND “Cognitive training” [All Fields] searched in the electronic databases PubMed and Embase and the Cochrane Library, with temporal limit (from 1 January 2010 until 30 April 2022). The search was updated on 30 September 2022. Two independent reviewers extracted data (CLR, EDM), screened the articles for title/abstract and full-text, removed the duplicates, and screened the reference lists of eligible articles and relevant systematic reviews published on the topic, the grey literature was searched on ClinicalTrials.gov. The primary authors of the studies were contacted in case of the unavailability of the studies. In case of disagreements in the steps elicited above, those were solved by a third author (LDN or VDL). Each step of this systematic review was discussed and peer-reviewed by two researchers (PKK, VDL). As this is a scoping review, its protocol was not eligible for registration in PROSPERO. All the authors (psychiatrists, medical doctors, and psychiatric rehabilitation technicians) are experts in the field of the contents and/or methods of the review.

### 2.2. Eligibility Criteria and Data Extraction

We included only randomized controlled clinical trials that met the PICO model identified for this study and the eligibility criteria: studies had to be (1) randomized controlled clinical trials; (2) conducted from 2010 onwards; (3) reported in English languages, and (4) focused on the use of the cognitive remediation in fully immersive VR program for the treatment of psychosocial disabilities (mental disorders). The eligibility criteria were applied independently by the primary authors of this study (PA, MGC, PrA), with any disagreement being solved by a third author (VDL or LDN). Two independent reviewers (CLR, EDM) extracted the following data via an ad hoc data extraction form: author, location; sample size/type of study, type of sample, control, type of intervention, measurement, fully immersive VR, duration of the intervention, dropouts, main findings. Were excluded: (1) duplicates; (2) argument not relevant (not mental disorders, not cognitive remediation program, not fully immersive virtual reality program); (3) not available after that we contact the author for the study request; (4) not RCT. A summary of the search strategy is shown in Table 1.

### 2.3. Quality Assessment of the Studies

To assess the risk of bias in the studies, we used the checklist for quality assessment of controlled intervention studies that was validated by the US National Heart Lung and Blood Institute [63]. For each item, a low risk of bias was assigned when the study met the expected criterion; a high risk of bias was assigned when the study did not meet the expected criterion; some concerns of bias were rated when the study did not report information about the criterion, or we cannot determine whether the criterion was met. Overall, a high quality of the study means that the majority of criteria met little or no risk of bias; an acceptable quality means that some criteria had some flaws in the study with some concerns for risk of bias; a low quality means that most criteria had significant flaws relating to key aspects of study design. The graphic presentation of the risk-of-bias assessment summary plot was created with the “robvis” package running in R [64].

### 2.4. Meta Analysis

Studies that were enough homogeneous in design were included in a meta-analysis. Overall, 8 studies on fully immersive VR for CR in subjects with MCI were surveyed. Effectiveness was estimated by comparing baseline data with data at the end of treatment [65]. The effect size was expressed as the bias-corrected standardized mean change score (Hedges’ g) and computed so that a positive value indicated a favorable outcome (e.g., improvement in cognition) [66]. More specifically, a positive effect size implicated that change in scores was greater in the treated than in the control condition. According to Cohen’s rule-of-thumb, effect size was interpreted as small when around 0.20; moderate when around 0.50; and large when ≥0.80 [67].

When a study included more than one measure for the same outcome, all relevant measures’ effect sizes were aggregated in a single score considering the measures correlations. If this information was not reported, a default correlation between measures was set at 0.5 and dependent effect sizes were aggregated [68].

Heterogeneity was assessed with Cochran’s Q and I^2^ statistics [69]. Heterogeneity was deemed negligible when I^2^ < 30%; moderate for values between 30 and 60%; substantial for 75–100% values [70]. Egger’s regression test could not be used because studies were less than ten [71]. Thus, publication bias was evaluated by using the trim-and-fill procedure [72]. The trim-and-fill method assumes that the most extreme results are not published and recalculates the effect size by the imputation of missing studies to produce a symmetrical funnel plot.

The radial plot was used to assess model adequacy [73]. For each study, the observation of a large, standardized residual (above 2, as a rule of thumb) suggests that the study does not fit the assumed model (i.e., it may be an outlier).

The results of both fixed- and random-effects models were reported. Between studies variance and variance of the effect size parameters across the population were estimated with the τ^2^ statistics using the Empirical Bayes estimator, with Knapp and Hartung adjustment for random-effects model. We calculated the 95% CI for the heterogeneity using the Q-Profile method, to assess the extent and relevance of heterogeneity [74]. The significance level threshold was set at *p* < 0.05.

Meta-analysis was carried out with R (version 4.2.2) [75] using the following packages: ‘metafor’ (version 3.8-1), ‘meta’ (version 6.0-0) and ‘MAd’ (version 0.8-3).

## 3. Results

### 3.1. Search Results and Study Selection

The flow diagram (Figure 1) shows the selection and screening process of the included articles. The initial screened included 4905 studied from the database and 7 from manual/citation searching articles. Of these, 1834 duplicates as well as 553 articles from database searches and 6 articles from manual/citations searching were excluded because they did not match the inclusion criteria. After the title, abstract screening, and eligibility assessment, 11 studies that met the inclusion criteria were included.

### 3.2. Descriptions of Studies

Among the selected articles all involved fully immersive VR-CR programs. Nine studies included people with Mild Cognitive Impairment (MCI), seven of these were defined as Randomized Clinical Trials [76,77,78,79,80,81,82] and two of them as pilot studies [83,84]. One included people with Mild Dementia [85] and it is a pilot study, and one included people with Schizophrenic Disorder defines as a Controlled Clinical Trial [86]. Six studies showed statistical differences in clinical outcomes between the experimental and control group. Specifically, Kang et al. 2021 [77] for quality-of-life outcomes, for the cognitive outcomes (attention, memory and executive function, global cognition); Maeng et al., 2021 [78] for depressive symptoms, for cognitive outcomes (languages, memory, executive function); Liao et al., 2019 [79] and Thapa et al., 2020 [80] for cognitive outcome (executive function); Liao et al., 2020 [81] for the daily functioning outcome; and Hwang et al., 2017 [82] for cognitive outcomes (memory). Two studies showed a statistically significant difference before and post-treatment for the experimental group but not between groups on global cognition [83,86] and on attention [86]. Three studies did not show any difference between time and/or groups. Two of these involved people with MCI [76,84] and one involved people with mild dementia [85]. The majority of the studies used a VR-CR program that trained all cognitive domains in a mixed ecological scenario (house room and open space city) [77,80,81,83,84]; two studies that trained all cognitive domains in a unique ecological scenario (supermarket) [76,78]; one used trained executive function in a shop scenario [79]; one that trained attention function in an open space scenario [86]; and two did not specify the cognitive domains trained and the type of scenario [82,85]. Among the six studies that showed statistical differences between the experimental and control group after treatment [77,78,79,80,81,82], the intervention method used was an overage of 60 min session and an average of 7 weeks with 3 sessions per week. Among the other five studies, the intervention method used was an overage of 50 min session and an average of 6 weeks with almost two sessions per week [76,83,84,85,86]. Only Kwan et al., 2021 [83] specifies the framework for the development intervention. Appendix A show a synthesis of the characteristics of the included studies (Appendix A).

### 3.3. Assessment of the Risk of Bias

Studies suffered from some bias in several key aspects of study design. In particular, quite never the statistical power was enough to detect the expected differences or assure replication of the study; randomization was often poor; problems were detected in the percentage of dropouts at the end of the study and in the adherence to the treatment of participants. Overall, studies with a low risk of bias were about 25% of the total, all others were rated with some concerns of bias or of low quality (Figure 2).

### 3.4. Synthesis of the Meta-Analysis

Five main outcomes were reconstructed by the studies (Appendix A, for the outcomes measures): executive functions, measured with Controlled Oral Word Association Test (COWAT), Stroop Test, Trial Making Test (TMT), Praxis Test, Digit Symbol Substitution Test (DSST), Executive Interview (EXIT-25); attention, measured with Digit Span Test, TMT, Winsconsing Card Sorting Test (WCST); memory, measured with Shiraz Verbal Learning Test (SVLT), Word List Memory Test (WLM), Digit Span, Auditory Verbal Learning Test (AVLT), Virtual Supermarket Test (VST); language, measured with Boston Naming Test (BNT), Words FluencyTest (WVF), Auditory Verbal Learning Test (AVLT); global cognition, measured with Montreal Cognitive Assessment (MOCA), Mini-Mental State Examination (MMSE), Consortium to Establish a Registry for Alzheimer’s Disease Neuropsychological Battery (CERAD). The measures were assigned as declared by the authors in the studies. Appendix A summarizes the results of the meta-analysis, which is further detailed hereafter (Appendix A).

#### 3.4.1. Executive Functions

Treatment did not improve executive functions (Figure 3). No outlier was detected based on the radial plot, and just one study was added by the Trim-and-Fill procedure, with no impact on the estimated effect (Appendix A). Cochran’s Q test did not detect statistically significant heterogeneity, and heterogeneity was estimated negligible to moderate based on I^2^ (95%CI = 0% to 74%).

#### 3.4.2. Attention

Treatment did not improve attention (Figure 4). No outlier was detected based on the radial plot and no study was added by the Trim-and-Fill procedure (Appendix A). Cochran’s Q test did not detect statistically significant heterogeneity, and heterogeneity was estimated negligible to moderate based on I^2^ (95%CI = 0% to 74%).

#### 3.4.3. Memory

Treatment improved memory in both the fixed-effects and the random-effects model (Figure 5). Effect size ranged from small to moderate. No outlier was detected based on the radial plot and no study was added by the Trim-and-Fill procedure (Appendix A). Cochran’s Q test did not detect statistically significant heterogeneity, however, heterogeneity was estimated negligible to substantial based on I^2^ (95%CI = 0% to 79%).

#### 3.4.4. Language

Treatment improved language according to the results of the fixed-effects model but not based on the random-effects model (Figure 6). No outlier was detected based on the radial plot and no study was added by the Trim-and-Fill procedure (Appendix A). Cochran’s Q test did not detect statistically significant heterogeneity, however, heterogeneity based on I^2^ was moderate (48%), ranging from negligible to substantial (95%CI = 0% to 83%).

#### 3.4.5. Global Cognition

For global cognition, too, treatment resulted effective according to the results of the fixed-effects model but not based on the random-effects model (Figure 7). No outlier was detected based on the radial plot, and just one study was added by the Trim-and-Fill procedure, with no relevant impact on the estimated effect (Appendix A). Cochran’s Q test detected heterogeneity (Q = 11.02; df = 4; *p* = 0.03), and heterogeneity based on I^2^ was moderate (64%), ranging from negligible to substantial (95%CI = 4% to 86%).

## 4. Discussion

The present study, to our knowledge, is the first meta-analysis about the use of a fully immersive VR-CR program for people with psycho-social disabilities. The systematic review suggested that VR has a positive impact on cognitive and functional outcomes in people with mental disorders and neurodegenerative disorders. The studies included in the systematic review involved people with MCI, one with mild dementia, and one with schizophrenic disorder. In general there were fewer studies with fully immersive VR-CR programs that involved people with different mental disorders. The studies reported a statistically significant difference between groups in cognitive functions (attention, memory, executive function, languages, and global cognition) [77,78,79,80,82], in daily functioning [81], in depressive symptoms outcome [78], and in quality of life [77]. Compared with the previous systematic review and meta-analysis that used traditional CR methods and not fully immersive VR [87,88] our results suggested not only a cognitive functions improvement but also a general clinical improvement. The meta-analysis involved only people with MCI showed a positive effect on clinical cognitive outcomes specifically to memory, language, and global cognition. It was not possible to analyze other general clinical outcomes due to their heterogeneity. In our study we also investigate the gap between effectiveness on clinical outcomes and the methods used in CR programs with fully immersive VR. Mainly, a higher weekly frequency with long-term duration, an ecological scenario, and multiple cognitive tasks trained are associated with a better clinical outcome, but often the instrument is not clearly described. With short-term intervention the positive impact is related only to memory outcomes, whereas with long-term intervention it is related to different cognitive outcomes. The majority of the studies did not specify the framework for the development intervention which is the explanation of the coherence between the hypothesis/outcomes/methods and the games used in the VR program and the cognitive domains trained. This lack of information does not permit the creation of a standard operating procedure, useful for the reproducibility of the intervention and the creation of a golden standard. In line with other reviews [55,56,57] the results of this review showed a poor quality of the studies; for instance, in terms of sample size and risk of bias, there is a large variability of the used methods such as frequency and the type of cognitive tasks trained. It should be noted that this systematic scoping review included only fully immersive VR, while the precedent reviews included a range from not immersive to fully immersive. In recent years various grants have been awarded for the implementation of VR and the metaverse for health treatment purposes. Based on recent revisions, so far, there are no studies that allow indicating the metaverse as a useful space for the exploration of virtual reality applications [89]. It is therefore essential to increase the methodological quality of the studies in order to take advantage of the use of new technologies with different kinds of immersion experiences in the cognitive deficit treatment in mental health, limiting the disadvantages, and ensuring the skills learning for daily real life and the respect of health rights.

### 4.1. Implication for Research and Clinics

More RCTs that study the effectiveness of fully immersive VR-CR programs for people with different psycho-social disabilities are necessary. Moreover, better methodological quality should be pursued; and more details should be reported about the negative effect of the use of VR and for the reference framework for the development of a complex intervention. In particular, the studies should better explain the hypothesis/outcomes/methods (for each session) and offer a better description of the games used and the cognitive domains in order to guarantee the repeatability of the intervention. This is necessary for the creation of a future golden standard.

### 4.2. Strengths and Limitations

This study used a comprehensive search strategy that made possible the inclusion of several studies on CR in fully immersive VR. However, many studies had poor quality. In particular, the dropout rate is not often specified and there is a risk of bias for the statistical and sample power. More studies with adequate quality and including different clinical applications in different populations rather than only MCI, are needed.

## 5. Conclusions

Understanding effectiveness concerning the methods used is an important aspect to create in the future a golden standard method of fully immersive VR-CR program for mental health rehabilitation, and a robust framework for the development of the intervention is necessary to achieve the rehabilitative goal as first the generalization in the daily life of the performed tasks and also to the reproducibility of the intervention in the clinical services.

## Figures and Tables

**Figure 1 ijerph-20-01527-f001:**
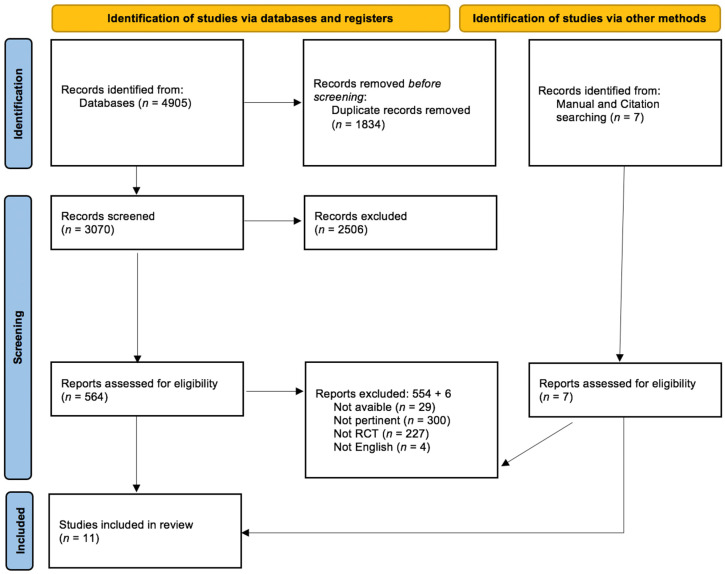
PRISMA 2020 flow-diagram.

**Figure 2 ijerph-20-01527-f002:**
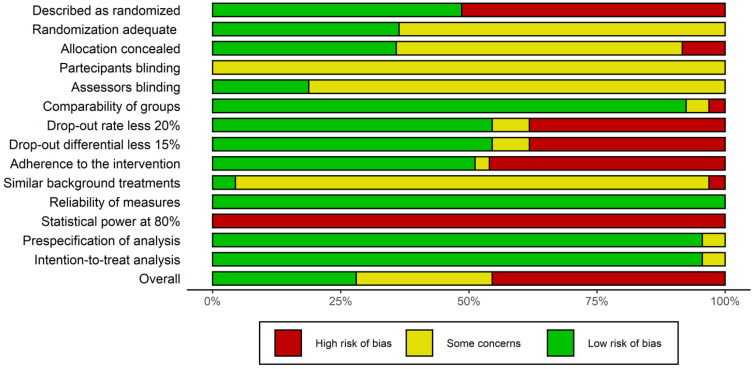
Risk of bias summary plot according to the National Institutes of Health Quality Assessment of Controlled Intervention Studies. Quality assessment of the included studies on cognitive remediation through fully immersive virtual reality in adults with psychosocial disabilities.

**Figure 3 ijerph-20-01527-f003:**
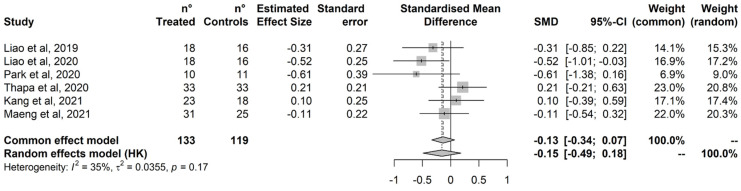
Executive Function. Liao et al., 2019 [79]; Liao et al., 2020 [81]; Park et al., 2020 [84]; Thapa et al., 2020 [80]; Kang et al., 2021 [77]; Maeng et al., 2021 [78].

**Figure 4 ijerph-20-01527-f004:**
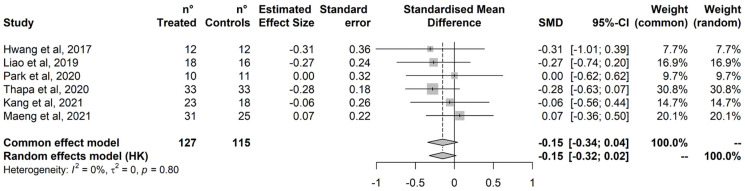
Attention. Hwang et al., 2017 [82]; Liao et al., 2019 [79]; Park et al., 2020 [84]; Thapa et al., 2020 [80]; Kang et al., 2021 [77]; Maeng et al., 2021 [78].

**Figure 5 ijerph-20-01527-f005:**
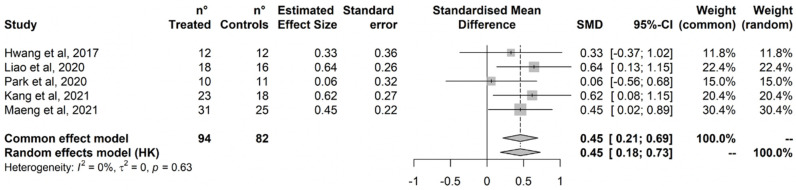
Memory. Hwang et al., 2017 [82]; Liao et al., 2020 [81]; Park et al., 2020 [84]; Kang et al., 2021 [77]; Maeng et al., 2021 [78].

**Figure 6 ijerph-20-01527-f006:**
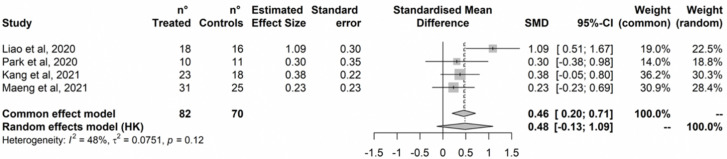
Language. Liao et al., 2020 [81]; Park et al., 2020 [84]; Kang et al., 2021 [77]; Maeng et al., 2021 [78].

**Figure 7 ijerph-20-01527-f007:**
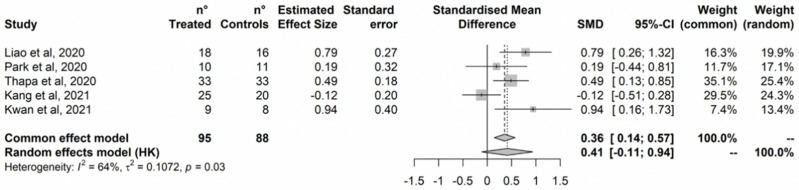
Global Cognition. Liao et al., 2020 [81]; Park et al., 2020 [84]; Thapa et al., 2020 [80]; Kang et al., 2021 [77]; Kwan et al., 2021 [83].

**Table 1 ijerph-20-01527-t001:** Summary of search strategy.

**Database searched**	Pubmed	
	Embase	
	Cochrane	
**Limits**	Language: English only.	
	Years: last 12 years (2022–2010)	
	Geographic: no limits	
**PICOS**	**Population**	People (adult, any gender) with psychosocial disabilities, specifically all the psychiatric diagnosis such as anxiety, psychotic, mood disorder, bipolar disorder etc. and neurocognitive disease such as mild cognitive impairment/Alzheimer’s disease and dementia
	**Intervention**	Studies that used cognitive remediation program in fully immersive virtual reality
	**Comparison**	No restrictions
	**Outcome**	Cognitive and clinical effectiveness, methods used
	**Study Type**	Only randomized clinical trials
**Exclusion**	Duplicates, not randomized clinical trials, articles not in English or not available, not cognitive remediation in fully immersive virtual reality program and not adult population with mental diseases	

## Data Availability

Data sharing not applicable.

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
