# Peer review of "Fully Immersive Virtual Reality-Based Cognitive Remediation for Adults with Psychosocial Disabilities: A Systematic Scoping Review of Methods Intervention Gaps and Meta-Analysis of Published Effectiveness Studies"

_ijerph, 2023, doi:10.3390/ijerph20021527_

Round 1

Reviewer 1 Report

All in all, the paper is well-written and add more evidence to previous findings. It is methodologically sound and findings are well presented.

There are some minor issues the authors may wish to address.

According to the study PICO, the interested population is individuals with psychosocial disabilities. Nonetheless, it seems this criterium is mostly inferred from diagnoses and not extracted per se. It would be then more accurate to use specific diagnoses to specify the targeted population.

Furthermore, a specific temporal limit (from January 1, 2010 to April 30, 2022) was decided. Is there a reason for reviewing the last 12 years? Please, specify that.

There are minor editing issues that can be easily corrected. MCI on page 2 is not initially spell out. There is a typo in table 1 languagge.

Author Response

Dear Reviewer,

Thank you very much indeed for the kind consideration and for allowing us to improve the manuscript according to the comments of the reviewers. Our replies to comments are listed below. Changes in the manuscript and the English revision were marked with “track changes” function.

All in all, the paper is well-written and add more evidence to previous findings. It is methodologically sound and findings are well presented. There are some minor issues the authors may wish to address.

Thank you so much for the positive comments, we really appreciated it.

According to the study PICO, the interested population is individuals with psychosocial disabilities. Nonetheless, it seems this criterium is mostly inferred from diagnoses and not extracted per se. It would be then more accurate to use specific diagnoses to specify the targeted population.

Thank you so much for the possibility to improve the PICOS. We wanted to include all types of psychiatric diagnoses and the main neurocognitive disease (Alzheimer's disease and Mild cognitive impairment), but we used the terms psychosocial disabilities in line with the new inclusive language according to the Convention on Rights of People with disabilities.

Furthermore, a specific temporal limit (from January 1, 2010 to April 30, 2022) was decided. Is there a reason for reviewing the last 12 years? Please, specify that. 

Thank you for this important consideration. We conducted this review before the pandemic period (we considered 10 years), but we decided to update with 2 years more. Before there weren’t studies with the use of fully immersive virtual reality, the majority included computerized cognitive training. In the last time, the use of virtual immersion is increasing.

There are minor editing issues that can be easily corrected. MCI on page 2 is not initially spell out. There is a typo in table 1 language.

Thank you for the advice, we corrected it.

Reviewer 2 Report

Thank you for the opportunity to review this interesting and innovative manuscript. It is well written and presented and the methodology is strong and complete. The manuscript is of interest for the community, it was systematically reviewed the literature and a meta-analyses was also performed. Virtual reality was analysed as a tool for the cognitive remediation in people with cognitive impairments. There are some major issues that have be solved before I can suggest this manuscript for publication:

-between the objectives it was stated that it will be presented a map of the currently published CR/VR interventions and the the methods components, evidence base… It should be appreciated a narrative analysis of the manuscript trying to extrapolate the common intervention elements and to present these aspects. Furthermore, a final summary, better if presented in a form of a “standard operating procedure” (recent review articles were published explaining its importance to improve the quality of the research in a area) of the “ideal intervention” should be appreciated to provide to future researchers a tool to adopt to follow the same direction.

-An implementation of the discussion is appreciated. I suggest to consider also the presentation of the metaverse as a tool to reach people everywhere and in anytime, a recent review on this topic is intitled “The metaverse: A new challenge for the healthcare system: A scoping review”

Minor comments:

Abstract: before to present the terms in their abbreviated form, please, associate them with the complete term. Cognitive Remediation (CR).

I suggest to further update the temporal limit to December.

Table 1. Please, correct english in English; correct rct in randomized controlled trials

Figure 1: please format the imagine, there are graphical errors.

Line 236-242: this paragraph is plenty of abbreviations such as COWAT, TMT… Please, present them in their extended version also in the text

Author Response

Dear Reviewer,

Thank you very much indeed for the kind consideration and for allowing us to improve the manuscript according to the comments of the reviewers. Our replies to the comments are listed below. Changes in the manuscript and the English revision were marked with “track changes” function.

Thank you for the opportunity to review this interesting and innovative manuscript. It is well written and presented and the methodology is strong and complete. The manuscript is of interest for the community, it was systematically reviewed the literature and a meta-analyses was also performed. Virtual reality was analyzed as a tool for the cognitive remediation in people with cognitive impairments. There are some major issues that have be solved before I can suggest this manuscript for publication: -between the objectives it was stated that it will be presented a map of the currently published CR/VR interventions and the the methods components, evidence base… It should be appreciated a narrative analysis of the manuscript trying to extrapolate the common intervention elements and to present these aspects. Furthermore, a final summary, better if presented in a form of a “standard operating procedure” (recent review articles were published explaining its importance to improve the quality of the research in a area) of the “ideal intervention” should be appreciated to provide to future researchers a tool to adopt to follow the same direction.

Thank you so much for this important comment that permitted us to better specify in the discussion section this aspect. Unfortunately, the lack of information did not permit us the definition of a standard operating procedure; specifically, we suggested the implementation of this information according to the objective reported in the comments.

-An implementation of the discussion is appreciated. I suggest to consider also the presentation of the metaverse as a tool to reach people everywhere and in anytime, a recent review on this topic is intitled “The metaverse: A new challenge for the healthcare system: A scoping review”

Thank you so much for this comment; we add in the discussion a specific consideration about the metaverse according to this review.

Minor comments:

Abstract: before to present the terms in their abbreviated form, please, associate them with the complete term. Cognitive Remediation (CR).

Thank you so much for the advice, we corrected it

I suggest to further update the temporal limit to December.

Thank you so much for the suggestion, we tried to update to December 2022 but we didn’t’ find new studies. 

Table 1. Please, correct english in English; correct rct in randomized controlled trials

Thank you so much for the correction, we corrected it.

Figure 1: please format the imagine, there are graphical errors.

Thank you. We tried to send also in pdf form because in our version we cannot find.

Line 236-242: this paragraph is plenty of abbreviations such as COWAT, TMT… Please, present them in their extended version also in the text

Thank you for the comment, which permits us to create coherence with the abbreviation and the complete term for the cognitive evaluation tests reported also in Table 2.

Round 2

Reviewer 2 Report

Thanks for the reply and corrections. The manuscript Is improved.